# Identification of cryptic subunits from an apicomplexan ATP synthase

**Diego Huet[1], Esther Rajendran[2], Giel G van Dooren[2], Sebastian Lourido[1,3]\***

[1]Whitehead Institute for Biomedical Research, Cambridge, United States; [2]Research School of Biology, Australian National University, Canberra, Australia; [3]Department of Biology, Massachusetts Institute of Technology, Cambridge, Massachusetts, United States

**Abstract** The mitochondrial ATP synthase is a macromolecular motor that uses the proton gradient to generate ATP. Proper ATP synthase function requires a stator linking the catalytic and rotary portions of the complex. However, sequence-based searches fail to identify genes encoding stator subunits in apicomplexan parasites like *Toxoplasma gondii* or the related organisms that cause malaria. Here, we identify 11 previously unknown subunits from the *Toxoplasma* ATP synthase, which lack homologs outside the phylum. Modeling suggests that two of them, ICAP2 and ICAP18, are distantly related to mammalian stator subunits. Our analysis shows that both proteins form part of the ATP synthase complex. Depletion of ICAP2 leads to aberrant mitochondrial morphology, decreased oxygen consumption, and disassembly of the complex, consistent with its role as an essential component of the *Toxoplasma* ATP synthase. Our findings highlight divergent features of the central metabolic machinery in apicomplexans, which may reveal new therapeutic opportunities.

DOI: https://doi.org/10.7554/eLife.38097.001

**\*For correspondence:**
lourido@wi.mit.edu

**Competing interests:** The authors declare that no competing interests exist.

## Introduction

The ATP synthase is a highly conserved protein complex found in the plasma membrane of bacteria, the inner membrane of mitochondria, and the thylakoid membrane of chloroplasts. The complex consists of two functionally distinct portions: the hydrophilic $F_1$ and the membrane-bound $F_o$ (*Walker, 2013*). The mechanism of this molecular motor is best understood for the mitochondrial ATP synthases of yeast and mammals. Within their mitochondria, the proton gradient generated by the electron transport chain (ETC) drives the rotation of a ring of $c$ subunits in $F_o$ and of the attached central stalk within $F_1$. This rotation causes the conformational changes in the $\alpha$ and $\beta$ subunits of $F_1$ that mediate catalysis of ATP from ADP and inorganic phosphate ($P_i$) (*Jonckheere et al., 2012*). The stator, also known as the lateral stalk, is an essential component of the ATP synthase because it counteracts the rotation of the $\alpha$ and $\beta$ subunits, enabling ATP synthesis (*Dickson et al., 2006*). It is therefore surprising that despite general conservation of the central subunits, the lateral elements of protozoan ATP synthases are structurally diverse, and these organisms appear to lack homologs for the stator subunits of yeast and mammals (*Lapaille et al., 2010*).

The composition of the mitochondrial ATP synthase has mainly been determined from detailed studies of purified *Saccharomyces cerevisiae* and *Bos taurus* mitochondria, both members of the eukaryotic clade Opisthokonta. In these two species, the architecture of the ATP synthase is virtually identical, and sequence analysis and proteomics have identified homology for nearly all of the subunits that constitute the mitochondrial ATP synthase (*Wittig and Schägger, 2008*). In contrast, recent studies in different protozoan species have reported unique structural and functional features in their ATP synthases. Proteomic and biochemical studies have identified several potential stator subunits in *Tetrahymena thermophila* and *Euglena gracilis*, although none appear conserved among protists

(*Balabaskaran Nina et al., 2010*; *Perez et al., 2014*). Similarly, the ATP synthases of algae like *Chlamydomonas reinhardtii* and *Polytomella* spp. and kinetoplastids like *Trypanosoma brucei* lack canonical stator subunits but possess phylum-specific proteins that likely perform that function (*Šubrtová et al., 2015*; *van Lis et al., 2007*).

Apicomplexans comprise a large phylum of obligate intracellular eukaryotic parasites that infect animals and cause significant human mortality and morbidity. Comparative genomics suggests that there is extensive metabolic diversity within this phylum, reflecting the variety of life cycles and host environments the various species have adapted to. Several core subunits of the ATP synthase (α, β, γ, δ, ε, c, and oligomycin sensitivity-conferring protein (OSCP)) can be found in most apicomplexan genomes (*Vaidya and Mather, 2009*). However, within the *Cryptosporidium* genus, drastic reductions in mitochondrial metabolism have coincided with preservation of only the α and β subunits in certain species (*Liu et al., 2016*; *Makiuchi and Nozaki, 2014*). Notably, the stator subunits and key components of the membrane-bound portion of this complex have not been identified in any apicomplexan species.

Reliance on mitochondrial metabolism can also vary over the course of a parasite's life cycle. For instance, the rapidly replicating forms of *T. gondii* catabolize glucose via the tricarboxylic acid (TCA) cycle (*MacRae et al., 2012*), whereas slow replicating stages seem to lack a functional TCA cycle and rely on glycolysis of stored amylopectin (*Denton et al., 1996*; *Uboldi et al., 2015*). A comparable metabolic shift seems to occur when *T. gondii* parasites egress from host cells and rely on oxidative phosphorylation for the majority of their ATP production (*Lin et al., 2011*; *MacRae et al., 2012*). *Plasmodium* spp. increase their TCA cycle activity as they transition from the asexual blood stages to the sexual cycle (*MacRae et al., 2013*). Consequently, deletion of the ATP synthase β subunit marginally affects blood stages but is essential for the sexual stages in the mosquito (*Sturm et al., 2015*). Together with pharmacological assays, these results confirm that the apicomplexan ATP synthase is active and essential in at least some life cycle stages (*Balabaskaran Nina et al., 2011*; *Sturm et al., 2015*). Direct measurements of ATP synthase activity in *T. gondii* (*Vercesi et al., 1998*) further suggest that these parasites must employ highly divergent stator subunits to form a functional enzyme.

To systematically examine gene function in *T. gondii*, our lab previously performed a CRISPR-based genome-wide screen that identified genes contributing to parasite fitness (*Sidik et al., 2016*). Approximately 360 of these fitness-conferring genes were broadly conserved among apicomplexans, but not other eukaryotes, and most lacked functional annotation; we referred to them as indispensable conserved apicomplexan proteins (ICAPs). Of the 17 ICAPs we determined a subcellular localization for, eight localized to the mitochondrion. This preponderance of ICAPs in the mitochondrion motivated further investigation.

Using unbiased proteomic and cryptic-homology analyses, we show that two mitochondrial ICAPs—ICAP2 and ICAP18—share homology with the ATP synthase stator subunits of opisthokonts. We show that both proteins interact with known and novel components of the ATP synthase, consistent with their role as putative subunits of the complex. Conditional depletion of ICAP2 leads to aberrant mitochondrial morphology and the disassembly of the ATP synthase complex, consistent with its proposed role in the stator. Our study reveals new information regarding the unconventional apicomplexan stator and provides a perspective on the divergent nature of the ATP synthase in this important group of pathogens.

## Results

### An indispensable conserved apicomplexan protein shares structural homology with bovine and fungal ATP synthase *b* subunits

ICAP2 (TGME49_231410) was originally described in our *T. gondii* genome-wide screens as one of the eight indispensable conserved apicomplexan proteins (ICAPs) that localized to the mitochondrion of the parasite (*Sidik et al., 2016*). These mitochondrial ICAPs lacked functional annotation, identifiable protein domains, or homology to sequences outside the apicomplexan phylum. We therefore used the protein function and structure prediction server HHPRED to search for homologs on the basis of profile hidden Markov models and secondary structure predictions (*Zimmermann et al., 2018*). Intriguingly, ICAP2 was predicted to contain a region of potential

homology to a stator component of a mammalian ATP synthase, the bovine *b* subunit (*Bt*ATPb; 97.8% probability), despite extremely low sequence identity between the two (~24%; *Figure 1A*). Subcellular localization and topology predictions support the presence of a mitochondrial targeting signal on ICAP2 as well as two transmembrane domains, similar to *Bt*ATPb. Despite the similarities, ICAP2 was predicted to be nearly twice as long as *Bt*ATPb. Unexpectedly, the analysis also predicted the presence of a putative calcium-binding domain in the N-terminal portion of ICAP2 (*Figure 1A*; 74.5% probability). ICAP2 homologs were present in most apicomplexan genomes with the exception of some *Cryptosporidium* spp. and *Gregarina niphandrodes*, which lack most known ATP synthase subunits (*Vaidya and Mather, 2009*). Homologs of ICAP2 were also found in *Chromera velia*, a free-living autotrophic relative of apicomplexans and *Perkinsus marinus*, a parasitic alveolate from the phylum Perkinzoa, but no additional related sequences were found outside the Alveolata. An alignment of the ICAP2 homologs shows extended regions of sequence identity including the putative calcium-binding domain and the region of similarity to *Bt*ATPb (*Figure 1B* and *Figure 1—figure supplement 1*). The alignment confirmed the extensive conservation of ICAP2 throughout the phylum (*Figure 1C*).

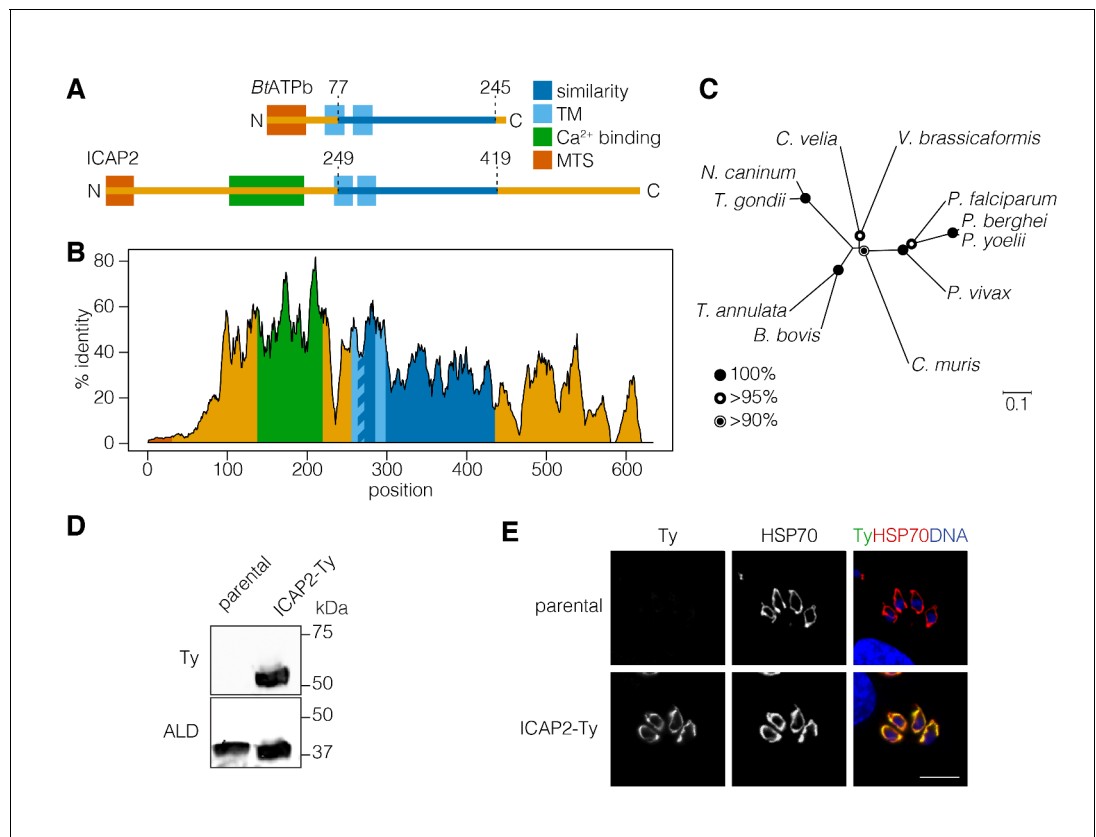

**Figure 1.** ICAP2 is a conserved apicomplexan protein with structural similarity to ATP synthase *b* subunits. (**A**) Schematic of the *Bos taurus* ATP synthase *b* subunit (*Bt*ATPb) and ICAP2 showing the mitochondrial targeting signal (MTS), transmembrane domains (TM), and the putative Ca²⁺-binding domain of ICAP2. The position of the region of similarity (dark blue) is numbered according to the amino acid sequence. (**B**) Percent identity plot of the aligned ICAP2 homologs from diverse apicomplexans (shown in *Figure 1—figure supplement 1*). Mean identity within a rolling window of ten residues is plotted. Domains are colored according to the schematic (**A**) following the positions of the *T. gondii* residues in the alignment. (**C**) Neighbor-joining tree showing the phylogenetic relationships of ICAP2 homologs analyzed in (**B**). Bootstrap values for 10,000 trials are displayed. (**D**) Immunoblot showing expression of Ty-tagged ICAP2 from its endogenous locus in the ICAP2-Ty strain. ALD serves as a loading control. (**E**) Intracellular parasites from the parental and ICAP2-Ty strains fixed and stained for Ty (green), HSP70 (red), and DAPI (blue). Scale bar is 10 μm.

DOI: https://doi.org/10.7554/eLife.38097.002

The following figure supplement is available for figure 1:

**Figure supplement 1.** Alignment of ICAP2 and its homologs in different apicomplexan and other alveolate species.

DOI: https://doi.org/10.7554/eLife.38097.003

Using CRISPR-mediated endogenous tagging, we introduced a C-terminal Ty epitope into the ICAP2 locus of the TATiΔ*KU80* strain of *T. gondii* (*Sheiner et al., 2011*). A clonal strain was generated by limiting dilution, and expression of ICAP2-Ty at the predicted molecular weight was confirmed by immunoblotting (*Figure 1D*). Localization of ICAP2 was determined by immunofluorescence where the protein colocalized with the mitochondrial marker HSP70 (*Pino et al., 2010*) (*Figure 1E*), in agreement with previous results (*Sidik et al., 2016*).

## ICAP2 associates with known and novel components of the ATP synthase

Based on the possible homology of ICAP2 to *Bt*ATPb and its mitochondrial localization, we sought to determine whether the protein interacts with the ATP synthase of *T. gondii*. We immunoprecipitated ICAP2 from the ICAP2-Ty strain using an antibody against Ty and then eluting the bound proteins under native conditions with excess Ty peptide. The parental strain was used as a specificity control. The input, unbound, and eluted fractions were resolved by SDS-PAGE and subjected to immunoblotting. This demonstrated that ICAP2 co-immunoprecipitated with the β subunit of the ATP synthase ($F_1\beta$), but not with an irrelevant cytosolic protein (ALD; *Figure 2A*). The total protein in the eluates was further visualized by silver stain, showing several proteins that co-immunoprecipitated with ICAP2-Ty (*Figure 2B*). We excised eight different sections of the gel and analyzed them by mass spectrometry. This analysis identified nearly all the annotated ATP synthase subunits found in *T. gondii* except for the *c* subunit, revealing that ICAP2 physically interacts with the ATP synthase complex.

Among the proteins immunoprecipitated with ICAP2 was TGME49_268830, which had been localized to the mitochondrion as part of a screen for apicoplast proteins (*Sheiner et al., 2011*). We referred to this protein as ICAP18 because it was part of our original ICAP list, although we were unable to tag it in the initial study (*Sidik et al., 2016*). HHPRED suggested that ICAP18 is homologous to the second major component of the bovine stator, the *d* subunit of the ATP synthase (*Bt*ATPd; 96.17% probability), despite only 17% sequence identity between the two proteins. Homologs of ICAP18 were present in apicomplexan and several other alveolate genomes, with the exception of organisms with reduced mitochondrial metabolism, mirroring the distribution of ICAP2. No additional sequences related to ICAP18 were found outside these groups. Similarity to *Bt*ATPd was restricted to the N terminus of ICAP18, where mitochondrial targeting signals and a transmembrane domain (TM) were predicted for both proteins (*Figure 2C*). ICAP18 additionally possesses a long C-terminal extension that contains a second predicted TM domain. All of these sequence features showed considerable conservation in the alignment of the apicomplexan and other alveolate ICAP18 homologs (*Figure 2D and E*, and *Figure 2—figure supplement 1*).

To further study the protein, we tagged the endogenous ICAP18 locus using CRISPR to introduce a Ty epitope tag by homologous recombination. A clonal tagged strain (ICAP18-Ty) was obtained by limiting dilution. Immunofluorescence confirmed the mitochondrial localization of ICAP18-Ty by colocalization with the HSP70 mitochondrial marker (*Figure 2F*). Taken together, these observations indicate that ICAP2 and ICAP18 are putative stator subunits of the *T. gondii* ATP synthase.

## A novel array of proteins is associated with the *T. gondii* ATP synthase

To directly test the association between ICAP18 and the ATP synthase, we performed an anti-Ty immunoprecipitation using ICAP18-Ty as the bait. This confirmed the interaction between ICAP18 and the β subunit of the ATP synthase ($F_1\beta$; *Figure 3A*). Comparison of eluates from the ICAP2-Ty and ICAP18-Ty strains showed comparable protein patterns when visualized by silver stain (*Figure 3B*). The lack of signal from the parental strain confirmed the specificity of immunoprecipitation.

To identify additional proteins associated with the *T. gondii* ATP synthase, we repeated the mass spectrometry analysis three times for ICAP2-Ty, and once for ICAP18-Ty, identifying a total of 209 proteins in aggregate (*Figure 3—figure supplement 1* and *Supplementary file 1*). To determine which proteins were most likely core components of the ATP synthase, we filtered the results based on several criteria (*Figure 3C*). First, we predicted that novel subunits would follow the same pattern of phylogenomic conservation as ICAP2, ICAP18, and most of the known ATP synthase subunits. We therefore performed BLAST (*Boratyn et al., 2013*) searches against each candidate to determine

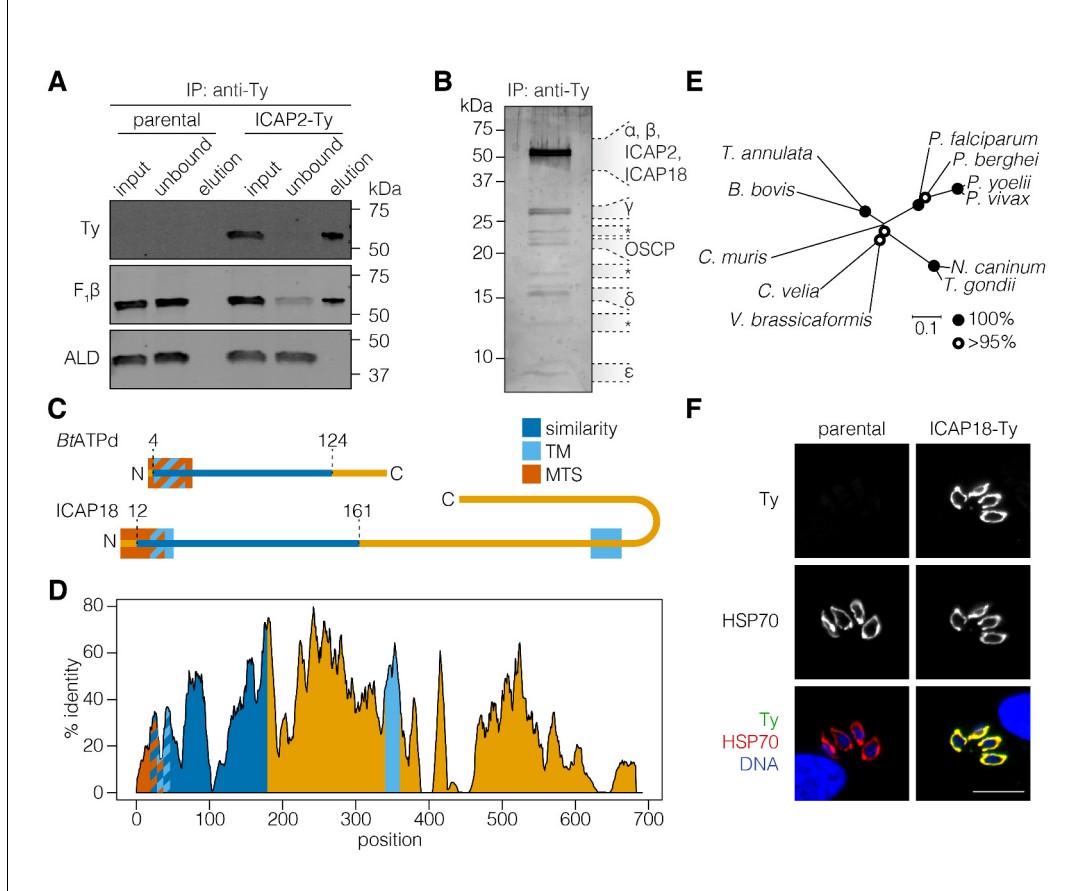

**Figure 2.** ICAP2 associates with known components of the ATP synthase. (**A**) Immunoprecipitation of Ty from the parental and ICAP2-Ty strains. Immunoblot for Ty, $F_1\beta$, and ALD in the input, unbound, and eluted fractions. (**B**) Following Ty immunoprecipitation, the ICAP2-Ty eluted fraction was separated by SDS-PAGE and proteins were visualized by silver staining. The eight gel fractions analyzed by mass spectrometry are labeled according to the known ATP synthase subunits identified in them, along with ICAP2 and ICAP18. Asterisks indicate bands where no known ATP synthase subunits could be identified. (**C**) Schematic representation of the *Bos taurus* ATP synthase *d* subunit (*Bt*ATPd) and ICAP18 showing the MTS, TMs, and the region of similarity (dark blue). (**D**) Percent identity plot of the aligned ICAP18 homologs from diverse apicomplexans (shown in Figure 2—figure supplement 1). Mean identity within a rolling window of ten residues is plotted. Domains are colored according to the diagram (**C**), and numbered according to the positions of the *T. gondii* residues in the alignment. (**E**) Neighbor-joining tree showing the phylogenetic relationships of the ICAP18 homologs analyzed in (**D**). Bootstrap values for 10,000 trials are displayed. (**F**) Intracellular parasites from parental and ICAP18-Ty strains fixed and stained for Ty (green), HSP70 (red), and DAPI (blue). Scale bar is 10 μm.
DOI: https://doi.org/10.7554/eLife.38097.004

The following figure supplement is available for figure 2:

**Figure supplement 1.** Alignment of ICAP18 and its homologs found in different apicomplexans and other alveolates.
DOI: https://doi.org/10.7554/eLife.38097.005

whether homologs could be found in *P. falciparum*, *C. velia,* and *C. muris* (E value $<10^{-5}$) but not *C. parvum* (E value $>10^{-5}$). Outside of the known subunits, only 16 proteins exhibited this pattern of conservation. The known ATP synthase subunits are also synchronously expressed along the cell cycle of *T. gondii* (*Behnke et al., 2010*). We therefore compared the expression pattern of each gene to the mean expression pattern of the known ATP synthase subunits by calculating the Spearman's correlation coefficient ($r_s$). Of the 16 genes above, nine were co-regulated with the known subunits ($r_s > 0.85$; *Figure 3D* and *Supplementary file 1*). All nine proteins, as well as all of the known subunits, were predicted to contribute to parasite fitness in our genome-wide screens (*Sidik et al., 2016*). The final list of novel subunits contained eight of the nine proteins. TGME49_226970 was excluded from the final list because it was clearly the ribosomal protein RPS11 (*Sun et al., 2015*). TGME49_215610 was ultimately included because it fit all the criteria except for coregulation, which could not be evaluated due to a lack of expression data. In addition to the stator

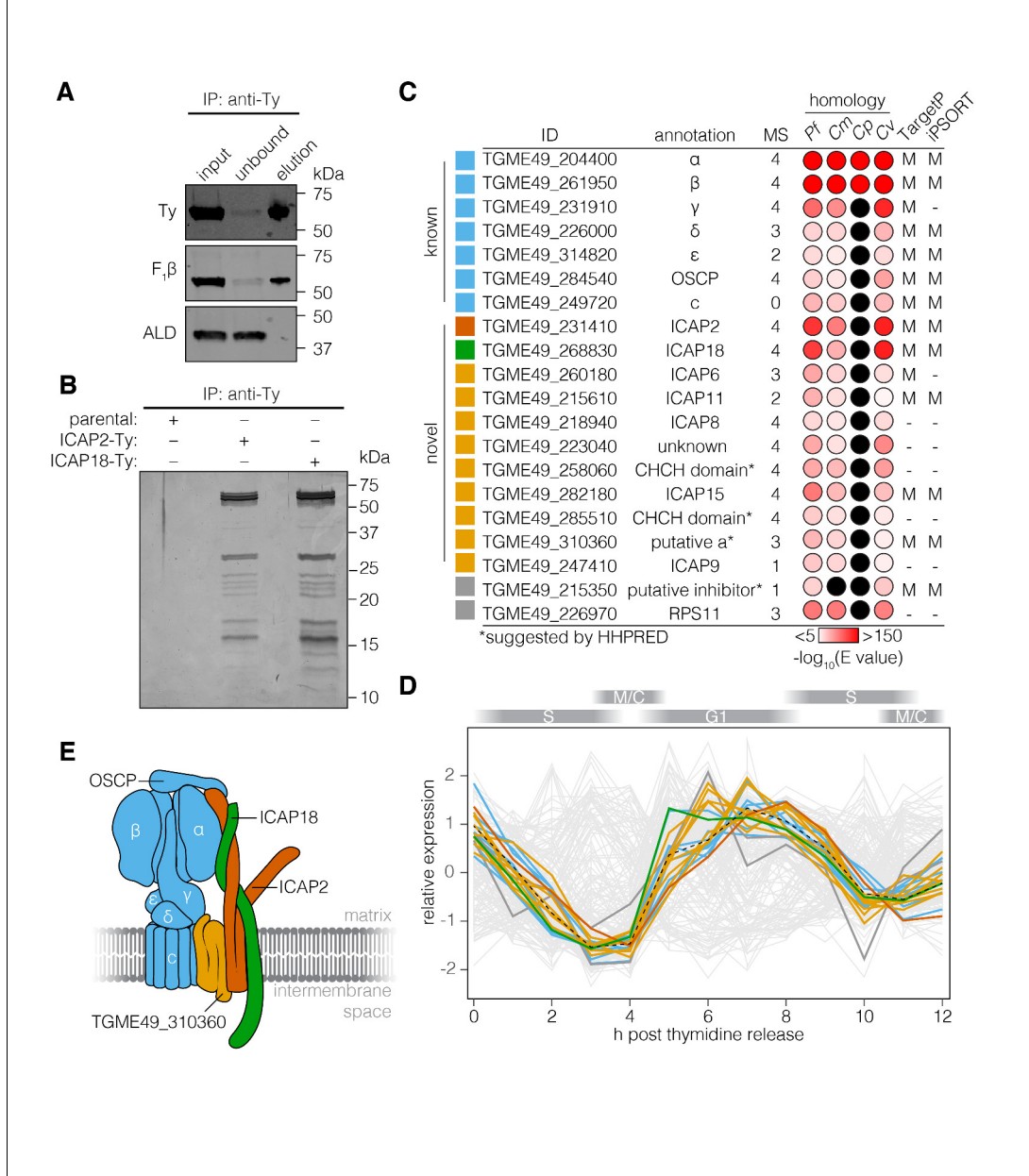

**Figure 3.** The *T. gondii* ATP synthase associates with proteins conserved among apicomplexans. (**A**) Immunoprecipitation of Ty from the ICAP18-Ty strain. Immunoblot for Ty, F₁β, and ALD in the input, unbound, and eluted fractions. (**B**) Following Ty immunoprecipitation, the eluates of the parental, ICAP2-Ty and ICAP18-Ty strains were separated by SDS-PAGE and visualized by silver staining. (**C**) Table showing the known and novel ATP synthase subunits, including ICAP2 (red) and ICAP18 (green). The table lists the proposed annotation of each gene and the number of times each protein was identified in the MS experiments. Proteins in grey represent candidates that did not meet the analysis criteria, or were clear contaminants. The homology indicates the BLAST expected value (E value) between each *T. gondii* protein sequence and that of its closest match in *Plasmodium falciparum* (*Pf*), *Cryptosporidium muris* (*Cm*), *Cryptosporidium parvum* (*Cp*), or *Chromera velia* (*Cv*). Cases in which a close match could not be identified (E value <0.0001) are indicated in black. The predicted subcellular localization ('M' for mitochondrial or '–' for another location) was determined using TargetP and iPSORT. (**D**) Relative expression pattern of known ATP synthase subunits (blue), ICAP2 (red), ICAP18 (green), and other novel associated proteins (yellow). Proteins that did not meet the analysis criteria are colored grey. The dotted line represents the mean relative expression of the known ATP synthase subunits. Cell-cycle stages are indicated above the plot. (**E**) Model of the ATP synthase including the known subunits (blue) and the predicted position of ICAP2 (red), ICAP18 (green), and the putative *a* subunit (orange). See also *Figure 3—source data 1*.
DOI: https://doi.org/10.7554/eLife.38097.006

The following source data and figure supplements are available for figure 3:

**Source data 1.** This file contains the source data used to make the graph presented in *Figure 3*.

*Figure 3 continued on next page*

*Figure 3 continued*

DOI: https://doi.org/10.7554/eLife.38097.009

**Figure supplement 1.** Co-immunoprecipitations of ICAP2 and ICAP18.

DOI: https://doi.org/10.7554/eLife.38097.007

**Figure supplement 2.** Alignment of putative *a* subunit and its homologs found in different apicomplexans and other alveolates.

DOI: https://doi.org/10.7554/eLife.38097.008

subunits, this list includes a putative *a* subunit adding to the components of the ATP synthase previously lacking from the annotation (*Figure 3E* and *Figure 3—figure supplement 2*).

Two proteins identified by mass spectrometry did not meet our criteria, but are nonetheless worth mentioning. TGME49_215350 was only identified in a single mass spectrometry experiment, and failed all of the criteria mentioned above, yet HHPRED suggested it is homologous to the yeast ATP synthase inhibitor (95.6% probability). In contrast, cytochrome $c_1$ (TGME49_246540), a component of complex III of the ETC, was identified in the four mass spectrometry experiments and met all of our criteria except for being marginally below the co-regulation cutoff.

All of the subunits in our high-confidence list were previously considered among the 360 indispensable conserved apicomplexan proteins (ICAPs) identified in our genome-wide screen. Five of these novel subunits (ICAP6, ICAP8, ICAP9, ICAP11, and ICAP15) were previously shown to localize to the mitochondrion (*Sidik et al., 2016*). When combined with the localization of ICAP2, six of the eight ICAPs previously localized to the mitochondrion are associated with the ATP synthase complex.

## ICAP2 is necessary for *T. gondii* growth

To further investigate the function of ICAP2, we generated a conditional knockdown strain (ICAP2-Ty cKD) using the recently developed U1-snRNP gene silencing strategy (*Pieperhoff et al., 2015*). This approach relies on the modification of the 3' end of a gene of interest in a DiCre-expressing strain to introduce a floxed synthetic 3' end followed by U1-binding sites. Activation of DiCre with rapamycin leads to recombination, which positions the U1 sites as part of the transcript and leads to mRNA degradation. Using CRISPR/Cas9, we modified the 3' end of ICAP2 accordingly, including an in-frame Ty tag and a hypoxanthine phosphoribosyltransferase (HXGPRT) selectable marker (*Figure 4A*). A 2 hr treatment of the cKD strain with rapamycin was sufficient to observe complete ICAP2-Ty degradation 48 hr later by immunoblot (*Figure 4B*) and immunofluorescence (*Figure 4C*).

To assess viability of parasites following ICAP2 depletion, we compared the number of plaques formed by parasites following a 2 hr rapamycin treatment to parasites treated with a vehicle control (DMSO). No plaques were observed upon ICAP2 depletion (*Figure 4D*) corroborating the protein's role in parasite fitness. We also measured intracellular growth by counting the number of parasites per vacuole at different time points following the rapamycin pulse. We consistently observed fewer large vacuoles in the rapamycin-treated cKD when compared to the paired controls, indicative of impaired cellular replication in ICAP2-depleted parasites (*Figure 4E*).

## Loss of ICAP2 triggers aberrant mitochondrial morphology

ATP synthase dimers and oligomers curve the mitochondrial inner membrane and contribute to the formation and maintenance of cristae (*Mannella, 2006*). In yeast, ATP synthase dysfunction has been linked to aberrant mitochondrial morphology, reduced cristae formation, matrix swelling, and organellar fragmentation (*Youle and van der Bliek, 2012*). We therefore tested the impact of ICAP2 depletion on mitochondrial morphology. Apicomplexan parasites possess a single mitochondrion that often encircles the nucleus of intracellular parasites (*Ovciarikova et al., 2017*). Immunofluorescence microscopy of ICAP2-depleted parasites indicated that mitochondria lose their characteristic lasso shape and appear to fragment 24–48 hr after rapamycin treatment (*Figure 5A*). To rule out any artifacts from fixation, we further investigated this phenotype by live microscopy using parental and ICAP2-Ty cKD strains expressing GFP fused to the mitochondrial targeting signal of SOD2 (SOD2-GFP), as previously described (*Pino et al., 2007*). Both strains also express the inner membrane complex protein IMC1 fused to TdTomato, allowing us to exclusively focus on non-dividing parasites and thereby exclude from our analysis the known mitochondrial dynamics that occur during

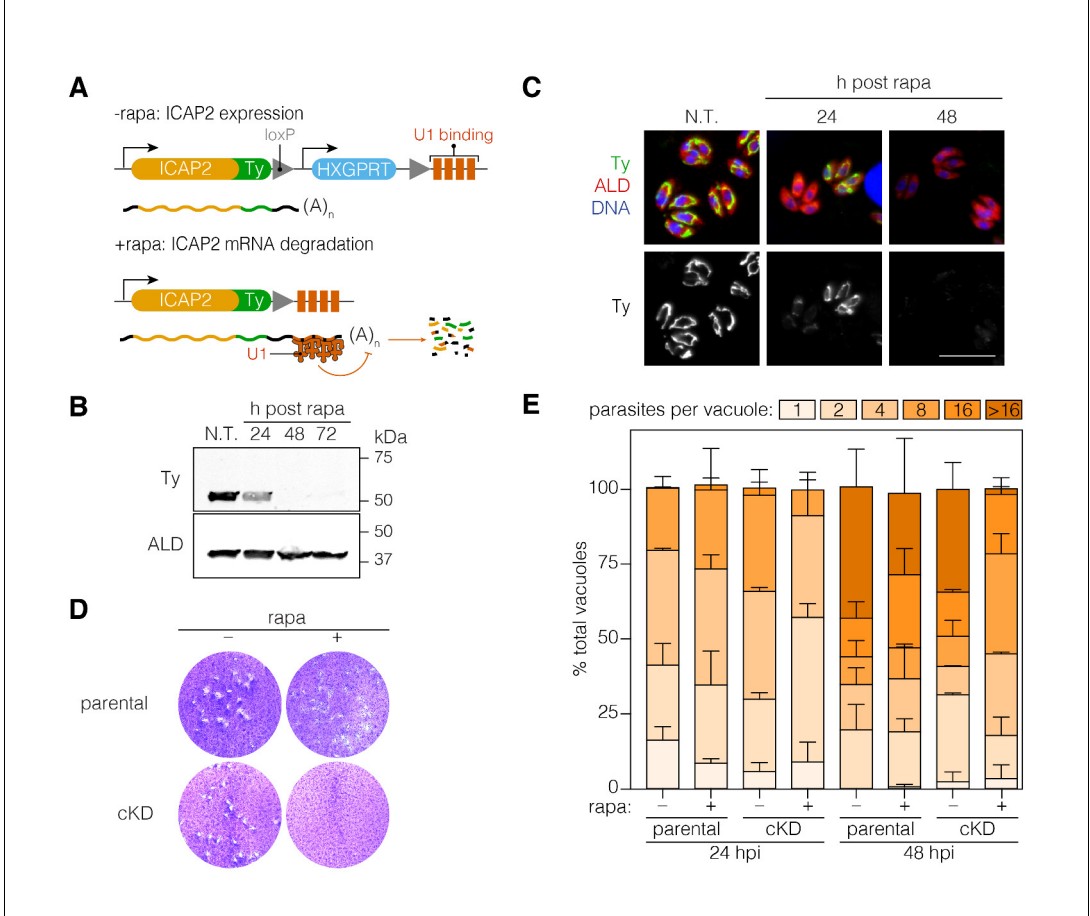

**Figure 4.** Functional characterization of ICAP2. (**A**) Schematic of the *ICAP2* locus in the ICAP2-Ty cKD strain indicating the U1-mediated mRNA degradation following the rearrangement caused by a brief pulse of rapamycin (rapa). (**B**) Immunoblot of the ICAP2-Ty cKD strain monitoring degradation of ICAP2-Ty at different time points following a 2 hr pulse with rapa. ALD serves as a loading control. (**C**) At different time points following treatment with rapa or vehicle (N.T.), intracellular ICAP2-Ty cKD parasites were fixed and stained for Ty (green), ALD (red), and DAPI (blue). Scale bar is 10 μm. (**D**) Plaque assay of the parental and ICAP2-Ty cKD strains after treatment with rapa or a vehicle control (DMSO). (**E**) The parental and ICAP2-Ty cKD strains were pulsed with rapa or a vehicle control 24 hr prior to passaging. Samples were fixed 24 or 48 hr post infection (hpi) and stained for Ty and ALD. The distribution of parasites per vacuole was determined. Bars represent mean ± SD for *n* = 2 independent biological replicates. At least 150 vacuoles were counted per condition in 2 or more technical replicates. See also *Figure 4—source data 1*.

DOI: https://doi.org/10.7554/eLife.38097.010

The following source data is available for figure 4:

**Source data 1.** This file contains the source data used to make the graph presented in *Figure 4*.

DOI: https://doi.org/10.7554/eLife.38097.011

the cell division (*Nishi et al., 2008*). We measured mitochondrial volume in vacuoles containing one or two parasites following treatment with rapamycin or a vehicle control (*Figure 5B*). Volume calculations were performed using Mitograph, an image processing method that enables estimates of the volume of three-dimensional organelles (*Viana et al., 2015*). In addition to the observed fragmentation, the mitochondrial volume of ICAP2-depleted parasites was significantly reduced at 24 and 48 hr after rapamycin treatment compared to untreated controls or the treated parental strain (*Figure 5C*).

Mutations in the *b* subunit of yeast ATP synthase lead to a loss of cristae and outer membrane involutions that give mitochondria an onion-like morphology (*Bornhövd et al., 2006*; *Weimann et al., 2008*). We therefore examined the ultra-structure of mitochondria in ICAP2-deficient parasites by transmission electron microscopy. Sections containing mitochondrial structures were blinded and randomized to measure mitochondrial area and quantify the number of cristae

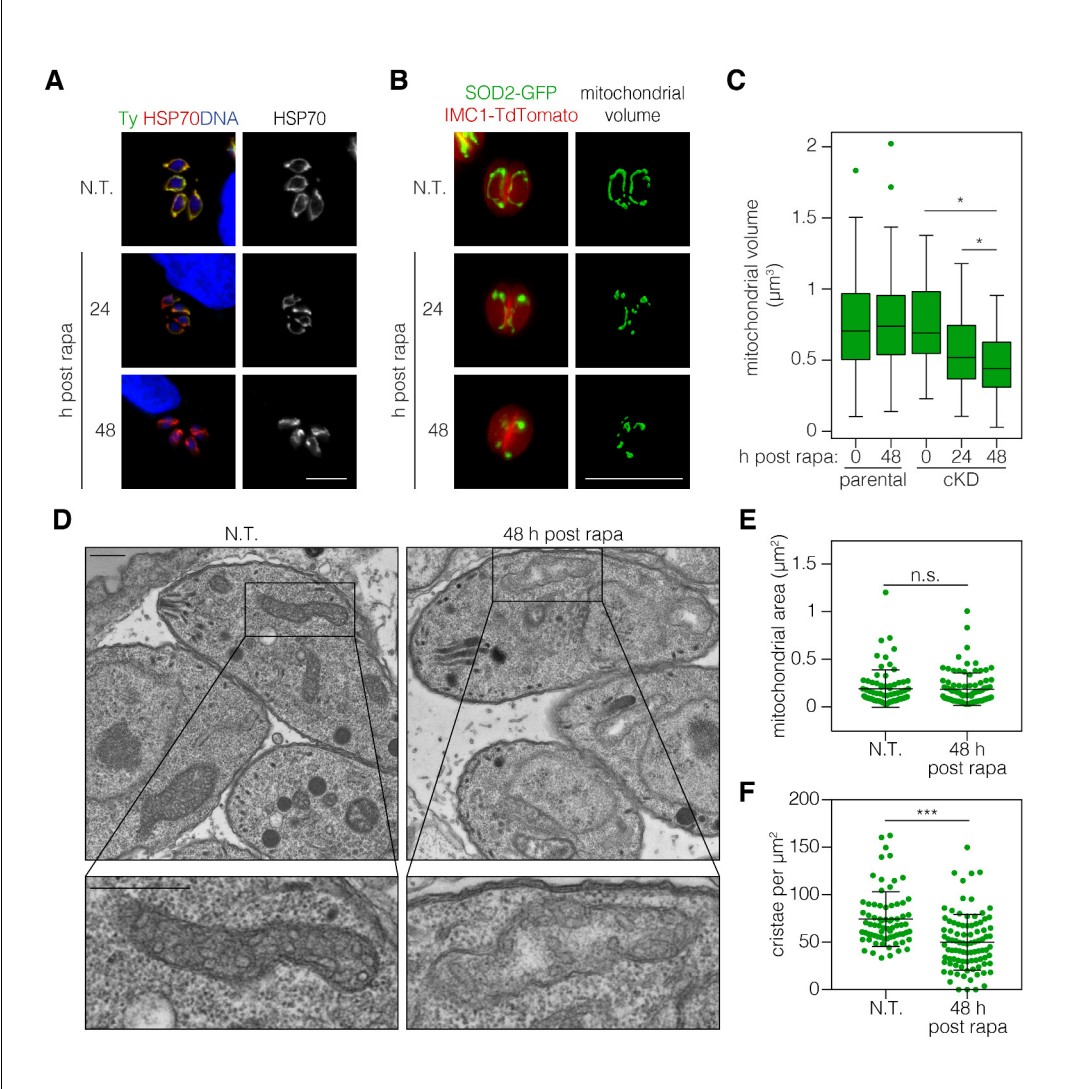

**Figure 5.** ICAP2 depletion alters mitochondrial morphology. (A) Mitochondrial morphology in the ICAP2-Ty cKD parasites visualized by staining for HSP70 (red), ICAP2-Ty (green), and DAPI (blue), at various time points following a 2 hr pulse with rapa. Scale bar is10 µm. (B–C) Changes in mitochondrial volume following ICAP2 depletion. Representative panels displaying the maximum intensity projections and mitochondrial volume from ICAP2-Ty cKD parasites expressing SOD2-GFP and IMC1-TdTomato (B). Live intracellular parasites were imaged following the various treatments. Scale bar is 10 µm. The experiment consisted of two biological replicates, and at least 100 vacuoles from three technical replicates were analyzed per condition (C). Boxplot, *p≤0.05; Student's *t*-test. (D–F) Electron micrographs of ICAP2-Ty cKD parasites 48 hr after a rapa pulse or treatment with a vehicle control. Insets show representative mitochondria. Scale bars are 500 nm. Blinded analysis of measured mitochondrial area and the number of cristae per µm² within those areas. Mean ± SD, n.s. p>0.05, ***p<0.0001 by a Mann-Whitney test. See also *Figure 5—source data 1*.

DOI: https://doi.org/10.7554/eLife.38097.012

The following source data is available for figure 5:

**Source data 1.** This file contains the source data used to make the graphs presented in *Figure 5*.
DOI: https://doi.org/10.7554/eLife.38097.013

(*Figure 5D–F*). We analyzed 73 sections for the control and 92 for the ICAP2-deficient parasites. The distribution of mitochondrial areas was comparable in the presence or absence of ICAP2, likely indicating similar thickness of the mitochondrial structures (*Figure 5E*). However, because the sections had been pre-selected to contain mitochondria, the similar areas do not reflect the general abundance or volume of the mitochondria within each sample. Notably, the number of cristae was significantly reduced upon ICAP2 depletion (*Figure 5F*), and in some cases the matrix appeared swollen.

Taken together, these results demonstrate the profound effects that ICAP2 loss has on the mitochondrial morphology of *T. gondii*.

## Loss of ICAP2 affects the function and conformation of the ATP synthase

The ATP synthase dissipates the proton gradient across the mitochondrial inner membrane, thereby increasing the activity of the ETC and leading to higher rates of oxygen consumption. To assess the effect of ICAP2 on ATP synthase activity, we measured the basal mitochondrial oxygen consumption rate (mOCR) in isolated parasites, following different periods of ICAP2 depletion. The basal mOCR of the mutant cells was significantly reduced 72 hr after rapamycin treatment (*Figure 6A*), indicating a reduction in ETC activity upon depletion of ICAP2. We simultaneously determined the extracellular acidification rate (ECAR), a measure of the parasite's metabolic activity that may include extracellular acidification caused by extrusion of lactate, a final product of glycolysis, and/or activity of the parasite proton-pumping ATPase (accompanying manuscript by Seidi et al.). ECAR was unchanged upon ICAP2 knockdown (*Figure 6—figure supplement 1*), suggesting that the defects we observed in mOCR are not caused by reduced parasite metabolic activity. The upper limit on mOCR imposed by proton transport through the ATP synthase can be relieved using carbonyl cyanide 4-(trifluoromethoxy)phenylhydrazone (FCCP), which acts as a mobile proton carrier that uncouples ETC function from ATP production. Treating cells with FCCP thereby allowed us to measure the maximum OCR from which we calculated the spare mOCR—the difference between basal mOCR (before FCCP addition) and maximum mOCR (after FCCP addition; *Figure 6—figure supplement 1*). We observed that the maximal mOCR was not significantly affected by ICAP2 depletion (*Figure 6—figure supplement 1*), and that spare mOCR was significantly increased (*Figure 6B*). These data are consistent with depletion of ICAP2 leading to defects in ATP synthase function, thereby impairing oxidative phosphorylation in the parasite.

To directly measure the role of ICAP2 in the activity of the ATP synthase, we measured ATP levels in freshly egressed parasites. Previous studies have demonstrated that extracellular parasites use glucose and glutamine—via glutaminolysis and a GABA shunt—as carbon sources for glycolysis and oxidative phosphorylation, respectively (*Lin et al., 2011*; *MacRae et al., 2012*). To suppress glycolysis, we incubated extracellular parasites with 2-deoxy-D-glucose (2-DG)—a competitive inhibitor of glycolysis. We further supplemented the media with glutamine or sufficient glucose to overcome 2-DG inhibition. ATP production was largely normal in strains lacking ICAP2 when glucose was used as the carbon source (*Figure 6C*). However, knockdown of ICAP2 significantly reduced ATP production when the carbon source was glutamine (*Figure 6C*). As previously demonstrated, ATP production from glutamine is oligomycin sensitive (*Lin et al., 2011*), and the remaining ATP generated from glutamine in knockdown parasites is likely the result of residual ATP synthase activity. These results demonstrate that ICAP2 is necessary for mitochondrial ATP production.

To determine whether ICAP2 plays a role in the stability of the ATP synthase complex—as would be expected of a stator subunit—we analyzed the abundance and migration of the β subunit using SDS and blue native (BN) polyacrylamide gel electrophoresis (BN-PAGE). The relative abundance of the β subunit did not vary in response to ICAP2 knockdown, as determined by immunoblot of lysates resolved by SDS-PAGE (*Figure 6D*). Immunoblots of the same lysates resolved by BN-PAGE showed the β subunit participates in a complex of approximately 900 kDa in untreated cells. However, upon ICAP2 depletion, the β subunit shifted to a much smaller complex, migrating at ~100 kDa (*Figure 6E*). A significant proportion of the total β subunit could be observed in this lower complex for the cKD as early as 24 hr after rapamycin treatment, and increasing thereafter. These observations are consistent with the dissolution of the ATP synthase complex upon loss of ICAP2, similar to what has been observed in yeast and mammalian mutants lacking stator subunits (*He et al., 2017*; *Kane et al., 2010*; *Soubannier et al., 2002*). Taken together, these results show that ICAP2 is required for the proper assembly and function of the *T. gondii* ATP synthase.

## Discussion

The elegant rotary mechanism of the mitochondrial ATP synthase requires a stator to impart the conformational changes on $F_1$ that drive ATP catalysis. It has therefore been a long-standing conundrum that most apicomplexan genomes appear to code for only the core subunits of the ATP synthase

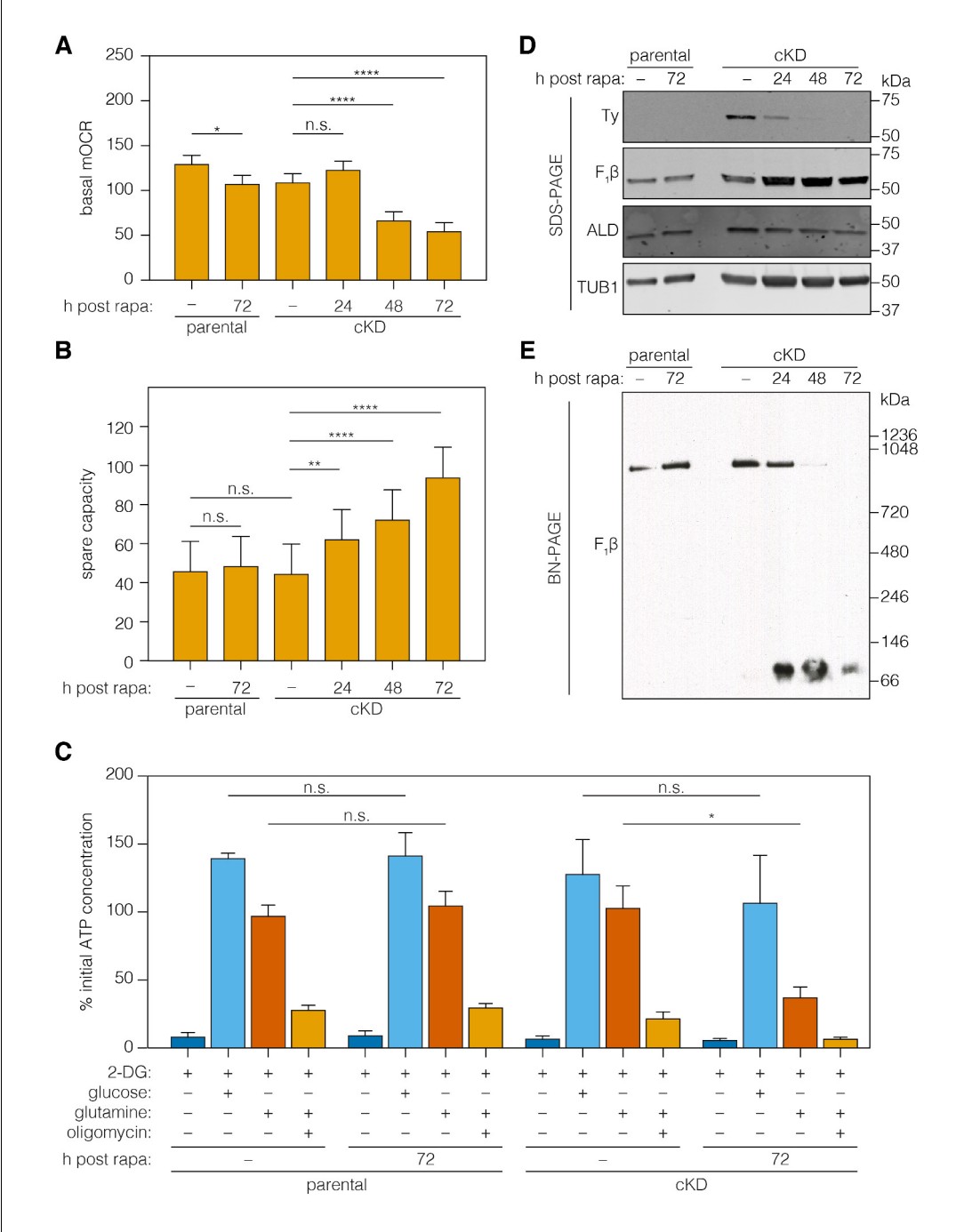

**Figure 6.** Loss of ICAP2 affects the function and integrity of ATP synthase. (**A–B**) Mitochondrial oxygen consumption (mOCR in pmol/min/1.5 × 10[6] parasites) was determined for the parental and ICAP2-Ty cKD strains at various time points following a 2 hr pulse with vehicle (–) or rapa. Basal mOCR (**A**) was compared to the maximum mOCR obtained after treating with the uncoupling agent FCCP to calculate the spare capacity (**B**). Data represent mean ± SEM for $n = 4$ independent experiments: **$p<0.005$; ****$p<0.0001$; n.s. not significant; one-way ANOVA followed by Tukey's test. (**C**) Relative ATP concentration for the parental and cKD strain cultured for 72 hr following a 2 hr pulse with vehicle (–) or rapa. ATP concentrations were measured following a 1 hr treatment with the indicated compounds and carbon sources and normalized to the initial ATP concentration of each strain (100%). Data represent mean ± SEM for $n = 3$ independent experiments for all treatments except glucose, which was only repeated twice; n.s. not significant; *$p≤0.05$; Student's $t$-test. (**D–E**) Lysates from the parental and ICAP2-Ty cKD strains were prepared at various time points following a 2 hr rapa pulse. Lysates were resolved by SDS-PAGE (**C**) or blue native PAGE (BN-PAGE) and blotted to probe for Ty, $F_1\beta$, ALD, or TUB1. See also *Figure 6—source data 1*.

DOI: https://doi.org/10.7554/eLife.38097.014

*Figure 6 continued*

The following source data and figure supplement are available for figure 6:

**Source data 1.** This file contains the source data used to make the graphs presented in *Figure 6*.
DOI: https://doi.org/10.7554/eLife.38097.016

**Figure supplement 1.** ECAR and OCR following downregulation of ICAP2.
DOI: https://doi.org/10.7554/eLife.38097.015

but none of the stator subunits. Here, we propose that ICAP2 (TGME49_231410) and ICAP18 (TGME49_268830) fulfill the roles of the stator *b* and *d* subunits, respectively. Both proteins localize to the mitochondrion of *T. gondii* and share distant homology with the respective subunits of the bovine ATP synthase, despite their unusual length and domain composition. Immunocapture of these putative subunits showed that they stably interact with the ATP synthase complex, along with at least nine other proteins that are conserved among apicomplexans. Moreover, conditional knock-down of ICAP2 leads to abnormal mitochondrial morphology, slow replication, and parasite death. ICAP2 was also necessary for the activity of the ATP synthase, and its absence caused a reduction in oxygen consumption by the mitochondrion and disassembly of the complex. Our results reveal highly divergent aspects of the apicomplexan ATP synthase and demonstrate their importance for parasite viability.

We previously identified ICAP2 as one of eight novel mitochondrial proteins, which we localized because of their roles in parasite fitness, a lack of functional domains, and conservation among api-complexans (*Sidik et al., 2016*). Pattern hidden Markov models and secondary structure predic-tions—implemented in HHPRED or Phyre2 (*Kelley et al., 2015*; *Zimmermann et al., 2018*)—first hinted at the homology between ICAP2 and the *b* subunit of the mammalian ATP synthase. These methods have also helped us assign putative functions to other subunits we isolated as part of the ATP synthase complex of *T. gondii*, including the putative *d* subunit (ICAP18; TGME49_268830), the putative *a* subunit (TGME49_310360), and the putative inhibitor (TGME49_215350). In opisthokonts, the *a* subunit is a multi-span membrane protein with a conserved arginine residue critical for its func-tion (*Mitome et al., 2010*). Topology predictions of the putative *a* subunit suggest the presence of three TM domains and an arginine residue that is conserved in apicomplexans and other alveolates (*Figure 3—figure supplement 2*). We have also identified other mitochondrial domains among the novel subunits. TGME49_258060 and TGME49_285510 bear homology to CHCH domain-containing proteins, which localize to the mitochondrial intermembrane space and play roles in biogenesis and bioenergetics (*Modjtahedi et al., 2016*). ICAP15 has a low-probability Bcl-like domain, common among proteins involved in apoptosis and regulation of the electron transport chain (*Gross and Katz, 2017*).

During the revision of this manuscript, a similar set of proteins was identified as part of a proteo-mic analysis of the purified *T. gondii* ATP synthase complex (*Salunke et al., 2018*). In addition to the novel subunits we described, the authors describe ten additional subunits, which our proteomic anal-ysis identified, but we excluded on the basis of lack of conservation in either *C. muris*, *P. falciparum*, or *C. velia* (*Supplementary file 1*). The novel subunits we propose may be closer to the core set of ATP synthase subunits present in alveolates; however, further studies will be needed to elucidate the function of most novel subunits regardless of their degree of conservation. ATP synthase com-plexes in other protists are characterized by a combination of conserved and phylum-specific subu-nits (*Balabaskaran Nina et al., 2010*; *Šubrtová et al., 2015*; *van Lis et al., 2007*). Our work demonstrates that distant homology searches can complement the molecular characterization of protein complexes in apicomplexans as in other divergent eukaryotes.

The length and extraneous domains present in ICAP2 make it a highly unusual *b* subunit, which has motivated the characterization we present. In particular, ICAP2 is nearly twice as long as its mammalian counterparts and appears to have an EF-like domain in its N terminus. EF domains bind cations like $Ca^{2+}$ or $Mg^{2+}$ and can act to modulate protein-protein interactions or enzymatic activity (*Kawasaki and Kretsinger, 2017*). Mitochondrial $Ca^{2+}$ uptake is known to regulate several meta-bolic processes (*De Stefani et al., 2016*), and the presence of an EF-like domain suggests that ATP synthase activity may be somehow coordinated with mitochondrial $Ca^{2+}$. Treatment of *T. gondii* with the ATP synthase inhibitor oligomycin causes an immediate increase in cytosolic $Ca^{2+}$ levels

(*Moreno and Zhong, 1996*). Although this may simply reflect the use of ATP by the endoplasmic-reticulum $Ca^{2+}$-ATPase, it also suggests a need for strong coupling between ATP production and $Ca^{2+}$ sequestration. Divergent stator paralogs are expressed in the *Drosophila* testis and underlie tissue specific properties of the ATP synthase that impact mitochondrial morphology and activity (*Sawyer et al., 2017*). Having provided strong evidence for the role of ICAP2 in the ATP synthase, more work will be needed to characterize the structure of the complex and understand how its unusual features mediate the specific adaptations of apicomplexan mitochondria.

ICAP2 is one of 11 novel subunits we believe are part of the apicomplexan ATP synthase. Orthologs of all of these subunits are found exclusively among alveolates, although they are absent from the genomes of species with reduced mitochondrial pathways. Several *Cryptosporidium* spp. (*C. hominis*, *C. parvum*, and *C. ubiquitum*) exhibit extreme reductions in the TCA cycle and electron transport chains, and concomitantly lack most of the known subunits of the ATP synthase along with the novel subunits we have identified (*Liu et al., 2016*). However, all *Cryptosporidium* spp. retain the α and β subunits, which can convert $ATP^{4-}$ to $ADP^{3-}$, fueling the generation of a membrane potential ($\Delta\psi$) across the inner mitochondrial membrane through the action of the ADP/ATP carrier (AAC). A similar mechanism has been proposed for the $F_o$-independent generation of $\Delta\psi$ in *Trypanosoma brucei* strains carrying mutations in the γ subunit (*Dean et al., 2013*). Even in the absence of an ATP synthase, $\Delta\psi$ is needed for the import of proteins into the matrix of mitochondria and mitosomes (*Fox, 2012*). Intriguingly, the genome of *Gregarina niphandrodes*, a divergent apicomplexan that infects beetles, lacks all of the mentioned ATP synthase subunits, including α and β along with homologs of the AAC (CryptoDB), and would require an alternative mechanism to generate $\Delta\psi$ if mitosomes are present and active in this organism.

We present the first direct evidence for the important role of the ATP synthase in *T. gondii* metabolism. Pharmacological and metabolomics approaches had previously demonstrated the presence of ATP synthase activity in *T. gondii* mitochondria and the importance of oxidative phosphorylation in supplying ATP to the parasite (*Lin et al., 2011*; *MacRae et al., 2012*; *Vercesi et al., 1998*). Consistent with these observations, depletion of ICAP2 reduces the rate of parasite replication and ultimately leads to a loss of parasite viability. This is in contrast to the mild effect of knocking out the β subunit in blood stages of *P. berghei*, although such mutants are subsequently impaired during the sexual stages of the life cycle (*Sturm et al., 2015*). The variable impact of the ATP synthase on the various stages of different apicomplexans may reflect the ability of these parasites to adapt to the changing environmental conditions they encounter across their life cycles.

ICAP2 depletion appears to damage *T. gondii* mitochondria leading to fragmentation and loss of volume. In other eukaryotes, mitochondrial fission segregates damaged parts of the organelle (*Youle and van der Bliek, 2012*). Mitochondrial fragmentation has been previously observed in *T. gondii* in response to exogenous stresses, including starvation (*Ghosh et al., 2012*) or monensin treatment (*Charvat and Arrizabalaga, 2016*). The effect of monensin was attributed to its uncoupling activity and to the induction of oxidative stress. Both effects could resemble defects in the ATP synthase like those predicted to result from loss of the stator; uncoupling would dissipate the $\Delta\psi$ used by the complex, and mutations in the $F_o$ subunits can lead to increased reactive oxygen species (*Baracca et al., 2007*). *Saccharomyces cerevisiae* stator subunit mutants have punctate, rounded mitochondria that display an altered ultrastructure (*Bornhövd et al., 2006*; *Weimann et al., 2008*). These defects result from the involvement of the stator subunits in cristae formation (*Cogliati et al., 2016*; *Wittig and Schägger, 2008*) and the uncoupling of ATP hydrolysis from proton translocation (*Paul et al., 1989*; *Soubannier et al., 2002*). Although ICAP2-depleted mitochondria did not show the smooth concentric membranes seen in *S. cerevisiae* stator mutants, they did have fewer cristae than wild-type mitochondria. To further understand the effects that lead to mitochondrial fragmentation, it will be interesting to compare uncoupling mutations, like ICAP2 loss, to disruption of the catalytic or proton-transporting functions of the *T. gondii* ATP synthase.

Without a functioning ATP synthase, the ETC is presumably inhibited by a buildup of $\Delta\psi$, which would normally be dissipated through the rotating $F_o$ (*Mueller, 2000*). We observed that the mOCR is reduced in ICAP2-depleted parasites and can be restored by the addition of a proton ionophore that dissipates $\Delta\psi$. In the accompanying paper by Seidi, Muellner-Wong, and Rajendran, et al., disruption of a novel apicomplexan component of the ETC complex IV also reduced the mOCR, although this effect could not be reversed by proton ionophore treatment. This indicates a general ETC defect in the complex IV mutant distinct from that observed upon ICAP2 depletion. Notably,

mutations in complex IV appear to have a milder effect on parasite viability and did not alter mitochondrial morphology. We also observed normal morphology upon ETC inhibition with atovaquone (a complex III inhibitor) and fragmented mitochondria following ATP synthase inhibition with oligomycin (data not shown).

The contribution of the mitochondrial ATP synthase to cellular ATP pools can be directly measured in *T. gondii* following repression of glycolysis by 2-DG and utilization of glutamine for oxidative phosphorylation (*Lin et al., 2011*). As expected, ICAP2-depletion severely compromised the production of ATP from glutamine, while its production from glycolysis remained largely unchanged. ICAP2 is therefore necessary for the proper function of the mitochondrial ATP synthase.

Together, these data indicate that the defects in mitochondrial morphology upon ICAP2 knockdown are not the result of general defects in the ETC, but rather a specific effect of ATP synthase impairment. In mammalian cells, inhibition of complexes I, III, or V (ATP synthase) resulted in distinct metabolic signatures (*Chen et al., 2016*). Similarly, in blood-stage malaria, it appears that the main purpose of the ETC is the regeneration of ubiquinone for pyrimidine biosynthesis (*Painter et al., 2007*). Therefore, inhibiting oxidative phosphorylation at different steps is expected to have different consequences, which should be investigated further as we explore these pathways as therapeutic targets in apicomplexans.

Our work identifies novel and highly divergent features of the apicomplexan ATP synthase, including two proteins that we believe comprise the complex's stator—ICAP2 and ICAP18. Previous studies had hypothesized the existence of a divergent stator in apicomplexans, and our observations confirm the presence and identity of such elements. We demonstrate the importance of these features by knocking down ICAP2, which leads to breakdown of the complex, defects in mitochondrial function and morphology, and loss of parasite viability. The conservation of these proteins in other apicomplexan species will also motivate their consideration as therapeutic targets. Future studies into the structure and function of these divergent features will help us understand their contributions to apicomplexan adaptation. Unlike most organisms, all of the ATP synthase subunits are encoded in the nuclear genome, making *T. gondii* a highly tractable organism to study the evolution of this important protein complex.

# Materials and methods

## Key resources table

| Reagent type (species) or resource | Designation | Source or reference | Identifiers | Additional information |
|---|---|---|---|---|
| Strain, strain background (Toxplasma gondii) | Parental (in *Figures 1*, *2* and *3*) | PMID: 22144892 | | TATiΔKU80 |
| Strain, strain background (T. gondii) | ICAP2-Ty | This paper | | TATiΔKU80 ICAP2-Ty |
| Strain, strain background (T. gondii) | ICAP18-Ty | This paper | | TATiΔKU80 ICAP18-Ty |
| Strain, strain background (T. gondii) | Parental in *Figures 4* and *5* and 6 | PMID: 26090798 | | DiCreΔKU80 |
| Strain, strain background (T. gondii) | ICAP2-Ty cKD | This paper | | DiCreΔKU80 ICAP2-Ty cKD |
| Strain, strain background (T. gondii) | Parental strain (expressing GFP fused to the mitochondrial targeting signal of SOD2 and expressing the A3:E11 membrane complex protein IMC1 fused to TdTomato) | This paper | | DiCreΔKU80 SOD2-GFP IMC1-TdT |
| Strain, strain background (T. gondii) | ICAP2-Ty (cKD strain expressing GFP fused to the mitochondrial targeting signal of SOD2 and expressing the A3:E11 membrane complex protein IMC1 fused to TdTomato) | This paper | | DiCreΔKU80 ICAP2-Ty cKD SOD2-GFP IMC1-TdT |

*Continued on next page*

*Continued*

| Reagent type (species) or resource | Designation | Source or reference | Identifiers | Additional information |
|---|---|---|---|---|
| Cell line (Homo sapiens) | Human Foreskin Fibroblasts (HFFs) | ATCC | SCRC-1041 | |
| Antibody | Mouse monoclonal anti-Ty1 (clone BB2) | PMID: 8813669 | | Dilutions: IFA: 1/1000, WB: 1/10000 |
| Antibody | Mouse monoclonal anti-TUB1 (clone 12G10) | Developmental Studies Hybridoma Bank at the University of Iowa | RRID: AB_1157911 | Dilution: WB 1/5000 |
| Antibody | Rabbit polyclonal anti-HSP70 | PMID: 17784785 | | Dilutions: IFA: 1/1000, WB: 1/10000 |
| Antibody | Rabbit polyclonal anti-ALD | PMID: 19380114 | | Dilutions: IFA: 1/1000, WB: 1/10000 |
| Antibody | Rabbit polyclonal anti-F1$\beta$ | Agrisera | Agrisera:AS05085 | Dilution: WB 1/5000 |
| Antibody | Goat anti-Mouse IgG (H + L) Secondary Antibody, DyLight 488 conjugate | Thermo Fisher | Thermo-Fisher:35502 | Dilutions: IFA: 1/1000 |
| Antibody | Goat anti-Rabbit IgG (H + L) Secondary Antibody, DyLight 594 conjugate | Thermo Fisher | Thermo-Fisher:35560 | Dilutions: IFA: 1/1000 |
| Antibody | Peroxidase AffiniPure Goat Anti-Rabbit IgG (H + L) | Jackson ImmunoResearch | Jackson ImmunoResearch:111-035-144 | Dilution: WB 1/10000 |
| Chemical compound, drug | Hoechst | Santa Cruz | Santa Cruz:sc-394039 | Dilutions: IFA: 1/20000 |
| Chemical compound, drug | Prolong Diamond | Thermo Fisher | Thermo-Fisher:P36965 | |
| Chemical compound, drug | Gentamicin | Thermo Fisher | Thermo-Fisher:15710072 | |
| Chemical compound, drug | Xanthine | Sigma-Aldrich | Sigma-Aldrich:X4002 | |
| Chemical compound, drug | Mycophenolic Acid | Sigma-Aldrich | Sigma-Aldrich:M3536 | |
| Chemical compound, drug | Rapamycin | EMD Millipore | EMD Millipore:553210 | |
| Chemical compound, drug | Carbonyl cyanide 4-(trifluoromethoxy) phenylhydrazone (FCCP) | Sigma-Aldrich | Sigma-Aldrich:C2920 | |
| Chemical compound, drug | Antimycin A | Sigma-Aldrich | Sigma-Aldrich:A8674 | |
| Chemical compound, drug | Atovaquone | Sigma-Aldrich | Sigma-Aldrich:A7986 | |
| Chemical compound, drug | Oligomycin | EMD Milipore | EMD Millipore:495455 | |
| Chemical compound, drug | 2-Deoxy-D-glucose (2-DG) | Sigma-Aldrich | Sigma-Aldrich:D6134 | |
| Chemical compound, drug | D-glucose | Thermo Fisher | Thermo-Fisher:15023021 | |
| Chemical compound, drug | Glutamine | Sigma-Aldrich | Sigma-Aldrich:G8540 | |
| Sequence-based reagent | All primers and oligonucleotides used in this study are listed in Supplementary file 2 | This paper | | |
| Recombinant DNA reagent | pU6-Universal | Addgene | Addgene:52694 | |
| Recombinant DNA reagent | pT8mycSOD2(SPTP) GFPmycHX | PMID: 17784785 | | |

*Continued on next page*

Continued

| Reagent type (species) or resource | Designation | Source or reference | Identifiers | Additional information |
|---|---|---|---|---|
| Recombinant DNA reagent | TubIMC1TdTomato-CAT | PMID: 26845335 | | |
| Recombinant DNA reagent | pG152-Lic-HA-FLAG-LoxP-3'UTRSag1-HXGPRT-LoxP-U1 | PMID: 26090798 | | |
| Recombinant DNA reagent | pG152-ICAP2-HA | This paper | | |
| Peptide, recombinant protein | Ty peptide | This paper | | |
| Commercial assay or kit | Gibson Assembly Cloning Kit | New England Biolabs | NEB:E5510S | |
| Commercial assay or kit | NucleoBond Xtra Midi | Macherey Nagel | Macherey Nagel:740412.50 | |
| Commercial assay or kit | CellTiter-Glo Luminescent Cell Viability Assay | Promega | Promega:15023021 | |
| Other | Pierce Protein G Magnetic Beads | Thermo Fisher | Thermo-Fisher:88847 | |
| Other | glucose and glutamine-free DMEM | Sigma-Aldrich | Silga-Aldrich:D5030 | |
| Other | FluoroBrite DMEM | Thermo Fisher | Thermo-Fisher:A1896701 | |
| Other | Halt protease inhibitor | Thermo Fisher | Thermo-Fisher:862209 | |
| Other | CellTak cell adhesive | In Vitro Technologies | In Vitro Technologies :FAL354240 | |
| Software, algorithm | HHPRED | PMID: 29258817 | | |
| Software, algorithm | ClustalX | PMID: 17846036 | | |
| Software, algorithm | Mascot, version 2.6.1 | Matrix Science | | |
| Software, algorithm | Scaffold, version 4.8.3 | Proteome Software | | |
| Software, algorithm | Prism, version 7 | Graphpad | | |
| Software, algorithm | R, version 3.2.3 | R Foundation for Statistical Computing | | |
| Software, algorithm | MitoGraph | PMID: 25640425 | | |
| Gene (T. gondii) | ICAP2 | N/A | ToxoDB:TGME49_231410 | |
| Gene (T. gondii) | ICAP18 | N/A | ToxoDB:TGME49_268830 | |

## Parasite culture and strains

*T. gondii* tachyzoites from the strain RH and its derivatives were maintained at 37°C with 5% $CO_2$ growing in human foreskin fibroblasts (HFFs) cultured in DMEM supplemented with 3% heat-inactivated fetal bovine serum and 10 μg/ml gentamicin. HFF cells were obtained from the ATCC (cat. no. SCRC-1041), authenticated by their standard procedures, and routinely tested negative for mycoplasma using the Universal Mycoplasma Detection Kit (ATCC, cat. no. 30–1012K). Parasites were transfected as described previously (*Sidik et al., 2014*). For selection, mycophenolic acid (Sigma, cat. no. M3536) and xanthine (Sigma, cat. no. X4002) were used at 50 μg/ml and 25 μg/ml, respectively.

## ICAP2 and ICAP18 phylogeny and topology predictions

ICAP2 and ICAP18 homologs were readily identified by BLAST searches against all sequenced api-complexan and chromerid genomes with the exception of *C. hominis*, *C. parvum*, and the gregarine *G. niphandrodes* (EupathDB.org). Alignment was performed using ClustalW (*Larkin et al., 2007*), and the phylogenetic tree was generated by neighbor-joining excluding positions with gaps. Boot-strap values were calculated for 10,000 trials. A hidden Markov model-based search of both proteins was performed for the alignment using HHpred (*Zimmermann et al., 2018*). Topology predictions were performed using CCTOP (*Dobson et al., 2015*), and mitochondrial targeting signal predictions were performed using iPSORT (*Bannai et al., 2002*). Subcellular localization predictions were performed with iPSORT and TargetP 1.1 (*Emanuelsson et al., 2000*).

## Plasmid construction

To generate the ICAP2-Ty cKD strain, a 1193 bp fragment from the 3' end of the *ICAP2* coding sequence was amplified for Gibson Assembly (New England Biolabs) from genomic DNA using P9 and P10. The pG152-Lic-HA-FLAG-LoxP-3'UTRSag1-HXGPRT-LoxP-U1 (a kind gift from Markus Meissner) was linearized with PacI, and the PCR product and PacI-linearized vector were Gibson cloned to generate the pG152-ICAP2-HA-FLAG-LoxP-3'UTRSag1-HXGPRT-LoxP-U1 plasmid. For simplicity, the plasmid will be referred to as pG152-ICAP2-HA. The HA tag present on this construct was then exchanged for a Ty tag by digesting it with EcoRI and PstI and inserting an in-frame Ty epitope (P11/P12) by Gibson Assembly, thereby generating the pG152-ICAP2-Ty vector. This vector was then used to generate a repair template containing the ICAP2-Ty-LoxP-3'UTRSag1-HXGPRT-LoxP-U1 cassette (referred to as ICAP2-Ty cKD cassette) flanked by homology regions to the C terminus of the *ICAP2* locus using P13 and P14.

## Strain generation

The ICAP2-Ty and ICAP18-Ty strains were generated by CRISPR-mediated C-terminal Ty tagging as previously described (*Sidik et al., 2016*). Briefly, 30 µg of a repair oligonucleotide, containing an in-frame Ty epitope flanked by homology arms targeting the C terminus of ICAP2 or ICAP18 (see *Supplementary file 2*), were co-transfected with 50 µg of pU6-Universal (Addgene, cat. no. 52694) carrying the appropriate sgRNA into TATiΔKU80 parasites (*Sheiner et al., 2011*). Transfected para-sites were cultured until their first lysis and used to infect confluent HFF monolayers grown on cover-slips. Correct expression and localization of the Ty-positive ICAPs was determined 24 hr post infection by immunofluorescence microscopy. Correct integration of the Ty epitope into the respective ICAP locus was confirmed by sequencing, and positive parasites were isolated and subcloned by limiting dilution. To generate the ICAP2-Ty cKD strain, the ICAP2-Ty cKD cassette was trans-fected into the DiCreΔKU80 strain along with the pU6-Universal plasmid carrying the same sgRNA that was used for endogenous tagging of ICAP2. Parasites were selected with xanthine and myco-phenolic acid, and correct expression and localization of the ICAP2-Ty positive parasites was deter-mined 24 hr post-infection by immunofluorescence microscopy. Finally, correct integration of the ICAP2-Ty cKD cassette into the ICAP2 locus was confirmed by sequencing, and parasites were iso-lated and subcloned by limiting dilution. For live microscopy experiments, the DiCreΔKU80 (*Andenmatten et al., 2013*) and ICAP2-Ty cKD strains were transfected with pT8mycSOD2(SPTP) GFPmycHX (*Pino et al., 2007*) and TubIMC1TdTomato-CAT plasmids. After their first lysis, dually fluorescent cells were FACS-sorted into 96-well plates to obtain single clones.

## Immunofluorescence microscopy and immunoblotting

Intracellular parasites were fixed with 4% formaldehyde at 4°C for 15 min and permeabilized with 0.25% Triton X-100 in PBS for 8 min. After blocking for 10 min with a PBS solution containing 5% normal goat serum (NGS) and 5% heat-inactivated fetal bovine serum (IFS), staining was performed with mouse anti-Ty (clone BB2) (*Bastin et al., 1996*), rabbit anti-ALD (a gift from L. David Sibley), or rabbit anti-HSP70 (a gift from Dominique Soldati-Favre) (*Pino et al., 2007*). Alexa-488-conjugated goat-anti-mouse (Invitrogen, cat. no. A11029) and Alexa-594-conjugated goat-anti-rabbit (Invitro-gen, cat. no. A11037) were used as secondary antibodies. Nuclei were stained with Hoechst (Santa Cruz, cat. no. sc-394039), and coverslips were mounted in Prolong Diamond (Thermo Fisher, cat. no. P36965). Images were acquired using an Eclipse Ti epifluorescence microscope (Nikon) using the

NIS elements imaging software. FIJI was used for image analysis, and Adobe Photoshop and Illustrator CC 2018 were used for image processing.

## Immunoblotting

For conventional immunoblotting, parasites were resuspended in lysis buffer (50 mM KCl, 20 mM HEPES pH 7.5, 2 mM $MgCl_2$, 0.1 mM EDTA, 1% Triton X-100, Halt protease inhibitors [Thermo Fisher, cat. no. 78440]). An equal volume of 2X Laemmli buffer (4% SDS, 20% glycerol, 5% 2-mercaptoethanol, 0.02% bromophenol blue, 120 mM Tris-HCl pH 6.8) was added, and the samples were heated to 100°C for 5 min prior to resolving by SDS-PAGE. After transferring the separated proteins to nitrocellulose, membranes were blotted with mouse anti-Ty or anti-TUB1 (clone 12G10, Developmental Studies Hybridoma Bank at the University of Iowa) and rabbit anti-ALD1, anti-HSP70, or anti-$F_1\beta$ (Agrisera, cat. no. AS05085). The signal was detected using 1:10,000 dilutions of IRDye 800CW-conjugated goat anti-mouse IgG and IRDye 680CW-conjugated donkey anti-rabbit IgG (LI-COR Biosciences) on an Odyssey infrared imager (LI-COR Biosciences). For BN-PAGE, parasites were solubilized with 4X Native PAGE Sample Buffer (Thermo Fisher Scientific) containing 2.5% digitonin. Separated proteins were transferred to a PVDF membrane and initially probed with mouse anti-$F_1\beta$ (MAb 5F4) (*Chen et al., 2015*) but the rabbit-anti-$F_1\beta$ from Agrisera was eventually used. In both cases, the membrane was incubated with a 1:10,000 dilution of horseradish peroxidase-conjugated goat-anti-rabbit antibody and developed with the ECL system (Amersham Biosciences) according to the manufacturer's instructions.

## Immunoprecipitations and silver stain

Prior to the immunoprecipitation, 60 µg of Ty antibody was cross-linked to 1 mg of Pierce Protein G Magnetic Beads (Thermo Fisher, cat. no. 88847). Approximately $2 \times 10^8$ parasites were resuspended in IP lysis buffer (150 mM NaCl, 20 mM Tris pH 7.5, 0.1%SDS, 1% Triton X-100, Halt protease inhibitors [Thermo Fisher]), incubated for 5 min at 4°C, and centrifuged at 21,000 x $g$ for 5 min. Supernatants were incubated with the Ty-magnetic beads for 1 hr at 4°C, and elution was carried out by adding a solution containing 150 ng/µl of Ty peptide diluted in IP lysis buffer to the beads, followed by a 30 min incubation at 4°C. Proteins were resolved by SDS-PAGE and visualized by silver stain as described in (*Shevchenko et al., 1996*).

## Plaque assays

To analyze the effect of ICAP2 on plaque formation, 500 DiCreΔ*KU80* or ICAP2-Ty cKD parasites per well were added to HFF monolayers in 6-well plates and treated with vehicle or rapamycin for 2 hr. All wells were washed twice with PBS, media was added, and eight days later the monolayers were rinsed with PBS, fixed in 95% ethanol for 10 min and stained with crystal violet (2% crystal violet, 0.8% ammonium oxalate, 20% ethanol) for 5 min.

## Live microscopy and mitochondrial volume measurements

Parental or ICAP2-Ty cKD parasites were treated with 50 nM rapamycin or vehicle (DMSO) upon infection of HFF monolayers in glass-bottom 35 mm dishes (MatTek). Rapamycin was removed after 2 hr, and parasites were allowed to grow for 24 or 48 hr. Vacuoles containing one or two parasites were imaged live using an Eclipse Ti microscope (Nikon) with an enclosure heated to 37°C and 5% $CO_2$. For each vacuole, a z-stack of approximately 25 frames at 0.2 µm spacing between each slice was taken. The bottom and top slices were set so that the mitochondria were just slightly out of focus. The mitochondrial volume was calculated using MitoGraph as described (*Viana et al., 2015*). The experiment consisted of two biological replicates, and at least 100 vacuoles from two technical replicates were analyzed per condition.

## Transmission electron microscopy

For ultrastructural analyses, infected HFF cells were fixed in a freshly prepared mixture of 1% glutaraldehyde (Polysciences Inc.) and 1% osmium tetroxide (Polysciences Inc.) in 50 mM phosphate buffer at 4°C for 30 min. The low osmolarity fixative was used to dilute soluble cytosolic proteins and enhance the visualization of mitochondria. Samples were then rinsed extensively in cold $dH_2O$ prior to en bloc staining with 1% aqueous uranyl acetate (Ted Pella Inc.) at 4°C for 3 hr. Following several

rinses in dH$_2$O, samples were dehydrated in a graded series of ethanol and embedded in Eponate 12 resin (Ted Pella Inc.). Sections of 95 nm were cut with a Leica Ultracut UCT ultramicrotome (Leica Microsystems Inc.), stained with uranyl acetate and lead citrate, and viewed on a JEOL 1200 EX transmission electron microscope (JEOL USA Inc.) equipped with an AMT 8-megapixel digital camera and AMT Image Capture Engine V602 software (Advanced Microscopy Techniques). For cristae enumeration, images were blinded and the mitochondrial area along with the corresponding number of cristae in each area were counted using FIJI. The images were subsequently un-blinded and the values for rapamycin-treated and vehicle were compared using a Mann-Whitney test.

## Mass spectrometry

After each immunoprecipitation experiment, bands from the SDS-PAGE gels were excised, divided into ~2 mm squares and washed overnight in 50% methanol in water. These were washed once more with 47.5% methanol 5% acetic acid in water for 2 hr, dehydrated with acetonitrile, and dried in a speed-vac. Reduction and alkylation of disulfide bonds was carried out by the addition of 30 µl 10 mM dithiothreitol (DTT) in 100 mM ammonium bicarbonate (NH$_4$HCO$_3$) for 30 min. Samples were alkylated by removal of the DTT solution and addition of 100 mM iodoacetamide in 100 mM NH$_4$HCO$_3$ for 30 min. Samples were sequentially washed with aliquots of acetonitrile, 100 mM NH$_4$HCO$_3$, and acetonitrile and dried in a speed-vac. The bands were enzymatically digested with 300 ng of sequencing-grade Trypsin (Promega, cat. no. V5111) in 50 mM NH$_4$HCO$_3$ for 10 min on ice. Sufficient NH$_4$HCO$_3$ was added to rehydrate the gel pieces. These were allowed to digest overnight at 37°C with gentle agitation. Peptides were extracted by gently agitating the gel bands 10 min with sequential 50 µl washes of 50 mM NH$_4$HCO$_3$ and twice with 47.5% acetonitrile 5% formic acid in water. All extractions were pooled in a 0.5 ml conical autosampler vial. Organic solvent was removed and the volumes were reduced to 15 µl using a speed-vac. Samples were analyzed by reversed-phase high-performance liquid chromatography using a Waters NanoAcquity in line with a ThermoFisher Orbitrap Elite mass spectrometer, in a nano-flow configuration. The mass spectrometer was operated in a top 10 data-dependent acquisition mode using a 240,000 FWHM m/z resolution in the Orbitrap and fragmentation in the linear ion trap.

## Mass spectrometry data analysis

Peptide searches were performed with Mascot (Matrix Science) algorithms against the *T. gondii* protein database (version 11, ToxoDB.org) concatenated to a database of common contaminants (keratin, trypsin, etc). The resulting Mascot search results were then loaded into Scaffold (Proteome Software), and a minimum peptide threshold of 95% was used for identification of peptides prior to the analysis described in the results. The proteomics data have been deposited to the ProteomeXchange Consortium via the PRIDE partner repository with the dataset identifier PXD009799.

## ECAR and mOCR measurements

Parental and ICAP2-Ty cKD parasites were incubated in 50 nM rapamycin or vehicle control for 2 hr, and cultured for a further 1–3 days before analysis. Freshly egressed parasites filtered through 3 µm were washed twice in Seahorse XF base medium (Agilent Technologies) supplemented with 1 mM L-glutamine and 5 mM D-glucose, before re-suspension to $1.5 \times 10^7$ parasites per ml. 100 µl of the parasite suspensions were seeded into wells of a 96-well culture plate coated with 3.5 µg/cm$^2$ of CellTak cell adhesive (In Vitro Technologies, cat. no. FAL354240). The plate was centrifuged at 50 x *g* for 3 min and an additional 75 µl of base medium was added to each well. Parasites were kept at 37°C in ambient air until the start of experiment. Parasite oxygen consumption rates (OCR) and extracellular acidification rate (ECAR) were measured using an Agilent Seahorse XFe96 Analyzer. To determine the maximal OCR, parasites were treated with FCCP to a final concentration of 1 µM. To determine the non-mitochondrial OCR, parasites were treated with 10 µM antimycin A and 1 µM atovaquone. OCR measurements were acquired for 3 min at 6 min intervals across the course of the experiment. Results were compiled using the Wave Desktop program, and analysis of parasite OCR was performed using the R software environment. Linear mixed-effects models were fitted to the data, with error between plates and wells (i.e. between and within experiments) defined as the random effect, and the OCR measurements in the different parasite strains and the time after rapamycin treatment defined as the fixed effect. A minimum of 4 wells was used for background correction

in each assay plate. Parasite mOCR was computed by subtracting the non-mitochondrial OCR measured following addition of antimycin A and atovaquone from the basal OCR. The spare capacity of the parasite ETC was computed by measuring the difference between basal mOCR and maximal mOCR (following the addition of FCCP).

## Measurement of cellular ATP concentration

Parental and ICAP2-Ty cKD parasites were grown in the presence or absence of 50 nM rapamycin for 2 hr, and subsequently cultured for 3 days before harvesting for the assay. Parasites were intracellular the day of the assay, and infected HFFs were washed three times with PBS to remove any extracellular parasites. Cells were then resuspended in FluoroBrite DMEM (Thermo Fisher, cat. no. A1896701) supplemented with 1% IFS and Halt protease inhibitor (Thermo Fisher, cat. no.1862209), scraped and syringe-released. Freshly released parasites were centrifuged at 80 x $g$ for 5 min to remove host cell debris, and the supernatant containing the parasites was removed and centrifuged at 400 x $g$ for 10 min and resuspended in glucose- and glutamine-free DMEM (Sigma, cat. no. D5030). Parasites were counted, washed once more in glucose- and glutamine-free DMEM, and resuspended at $6 \times 10^6$ parasites per ml. 50 µl aliquots of each strain were arrayed in a 96-well PCR plate, and 50 µl of the corresponding compounds diluted in glucose- and glutamine-free DMEM were added accordingly to each well. 2-DG (Sigma, cat. no. D6134) was used at 5 mM, glucose (Thermo Fisher, cat. no. 15023021) was used at 25 mM, glutamine (Sigma, cat. no. G8540) was used at 2 mM and oligomycin (EMD Milipore, cat. no. 495455) at 20 µM. Parasites were incubated with the respective treatments at 37°C, 5% $CO_2$ for an hour. To determine the initial ATP concentrations, control samples were immediately flash frozen and all other samples were flash frozen following incubation. To measure ATP concentration, 100 µl of CellTiter-Glo Reagent (Promega, cat. no. G7572) was added to the frozen samples, which were then allowed to thaw and equilibrate for 1 hr at room temperature. After equilibration, 100 µl from each well was transferred to a flat-bottom, black, 96-well plate. Luminescence was measured with an Envision microplate reader (Perkin-Elmer). For each strain and treatment, ATP levels were normalized to the initial ATP concentration.

## Acknowledgements

We thank Eric Spooner at the Whitehead Proteomics Core and Wandy Beatty at the Washington University Molecular Microbiology Imaging Facility for scientific services, L David Sibley for the ALD antibody, Peter Bradley for the $F_1\beta$ antibody, Dominique Soldati-Favre for the HSP70 antibody and the SOD2-GFP vector, and Markus Meissner for the DiCreΔKU80 strain and the U1-tagging plasmid. We would also like to thank Benedikt Markus and Emily Shortt for technical support and useful discussions. This work was supported by the NIH Director's Early Independence Award (1DP5OD017892) and an NIH Exploratory R21 grant (1R21AI123746) to SL, and an NIH Pathway to Independence Award (K99AI137218) to DH.

## Additional information

### Funding

| Funder | Grant reference number | Author |
| --- | --- | --- |
| National Institutes of Health | 1DP5OD017892 | Sebastian Lourido |
| National Institutes of Health | 1R21AI123746 | Sebastian Lourido |
| National Institutes of Health | K99AI137218 | Diego Huet |

The funders had no role in study design, data collection and interpretation, or the decision to submit the work for publication.

### Author contributions

Diego Huet, Conceptualization, Formal analysis, Investigation, Visualization, Writing—original draft; Esther Rajendran, Investigation, Methodology, Writing—review and editing; Giel G van Dooren, Funding acquisition, Project administration, Writing—review and editing; Sebastian Lourido,

Conceptualization, Data curation, Formal analysis, Supervision, Funding acquisition, Visualization, Writing—original draft, Writing—review and editing

### Author ORCIDs
Giel G van Dooren 🔟 https://orcid.org/0000-0003-2455-9821
Sebastian Lourido 🔟 http://orcid.org/0000-0002-5237-1095

### Decision letter and Author response
Decision letter https://doi.org/10.7554/eLife.38097.023
Author response https://doi.org/10.7554/eLife.38097.024

## Additional files

### Supplementary files
• Supplementary file 1. Summary of mass spectrometry data showing proteins identified in ICAP2-Ty and ICAP18-Ty immunoprecipitations.
DOI: https://doi.org/10.7554/eLife.38097.017
• Supplementary file 2. Primers and oligonucleotides used in this study.
DOI: https://doi.org/10.7554/eLife.38097.018
• Transparent reporting form
DOI: https://doi.org/10.7554/eLife.38097.019

### Data availability
Proteomics data have been deposited in ProteomeXchange Consortium via the PRIDE partner repository with the dataset identifier PXD009799 and 10.6019/PXD009799. All other data are included in the manuscript and supporting files.

The following dataset was generated:

| Author(s) | Year | Dataset title | Dataset URL | Database, license, and accessibility information |
|---|---|---|---|---|
| Diego Huet | 2018 | ICAP2 Immunoprecipitation results | http://central.proteomexchange.org/cgi/GetDataset?ID=PXD009799 | Publicly available at ProteomeXchange (accession no. PXD009799) |

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
