## [Decision Letter]

Thank you for submitting your article "Identification of cryptic stator subunits from an apicomplexan ATP synthase" for consideration by *eLife*. Your article has been reviewed by three peer reviewers, including Dominique Soldati-Favre as the Reviewing Editor and Reviewer #1, and the evaluation has been overseen Anna Akhmanova as the Senior Editor. The following individuals involved in review of your submission have agreed to reveal their identity: Martin Blume (Reviewer #2); Akhil Vaidya (Reviewer #3).

The reviewers have discussed the reviews with one another and the Reviewing Editor has drafted this decision to help you prepare a revised submission.

Summary:

Mitochondrial ATP synthase complex sits at the center of energy economy of most eukaryotic organisms. This nanomotor has highly conserved subunit composition and is made up of two sectors, F_1_, consisting of the catalytic subunits, and F_o_, consisting of subunits that constitute proton channels that drive rotary motion. Previous studies have shown the presence of core subunits that make up the F_1_ sector in many unicellular eukaryotes including most members of the phylum Apicomplexa. However, except for the *c* subunit, F_o_ sector subunits could not be detected through bioinformatic approaches in many of these organisms. The manuscript by Huet et al. reports the discovery of several such "missing" subunits in *Toxoplasma gondii* and conservation of these subunits in most apicomplexans and chromerids. These findings represent an important advance with implications for evolution of this central player in bioenergetics as well as for potential approaches to exploit the differences seen here to develop new therapeutic approaches. The work is carried out competently with a variety of approaches used to support their findings.

Essential revisions:

1) A manuscript currently deposited at bioRxiv (Salunke et al.; https://www.biorxiv.org/content/early/2018/05/14/321620 (now published in PLOS Biology, Salunke et al., 2018)) also describes subunit composition of *Toxoplasma* ATP synthase complex. In this manuscript, the authors find 20 novel subunits in addition to the 7 previously known subunits. In contrast, Huet et al. report just 11 novel subunits. Salunke et al. manuscript list 10 of the 11 subunits that Huet et al. report, which is quite reassuring, but the absence of 10 subunits listed by Salunke et al. in Huet et al. list is disconcerting. Huet et al. use an elaborate filtering approach to eliminate a large number of proteins detected through LC-MS/MS. The question thus arises as to whether the 10 additional subunits seen by Salunke et al. were filtered out by Huet et al. or that these were not detected in the first place. One possible reason for the discrepancy could be the tagged subunit used by the authors to pull down the complex for LC-MS/MS analysis: Huet et al. used the *b* subunit (present at 1:1 stoichiometry) whereas Salunke et al. used the β subunit (present at 3:1 stoichiometry). (Action to take: specify more clearly the stringency criteria applied to your analysis.)

2) The authors do not provide direct biochemical evidence for the proposed roles of ICAP2/18 in a functional ATP synthase complex. As proxy, oxygen consumption is measured in an ICAP2 knockdown mutant. This provides only an indirect measure as ATP synthase does not reduce oxygen. Further the seahorse data supporting this are incompletely reported as bar graphs.

Glycolytic and mitochondrial contributions to ATP production in extracellular tachyzoites was characterized before (Lin et al., 2011) and mitochondria were found to contribute a small but quantifiable fraction to cellular ATP. The assay and conditions used in that paper glucose-free, 2-DG and glutamine supplemented should enable a more direct measurement of ATP in the ICAP2 knockdown mutant. (Action to take: to be addressed experimentally.)

3) The authors are missing an important opportunity to point out implications of their findings in reference to the evolution of the apparent monophyletic clade Alveolata. The absence of the ATP synthase subunits they have found in Toxoplasma from the ciliates (and vice versa, i.e. ciliate ATP synthase subunits are absent in other alveolates) suggests almost a complete replacement of ATP synthase subunits at the branch point separating ciliates from the rest of the alveolate clade. Indeed, authors should include dinoflagellates in their comparative analysis (they actually have these "missing" subunits with orthology to *Toxoplasma*, as pointed out by Salunke et al.). The chromoalveolate hypothesis has suggested monophyletic origin of all organisms possessing secondary red algal plastid and implying that ciliates may have lost their plastids. There are other studies that argue against this, and thus the present study provides further evidence against chromoalveolate hypothesis. One possibility is that the novel ATP synthase subunits shared by apicomplexans, chromerids (and dinoflagellates) may have originated from the nuclearly encoded genes of the secondary endosymbiont that targeted them to its mitochondrion, which now having been lost, got incorporated in the host nucleus for its own mitochondrion. (Action to take: extend the Discussion section.)

4) Importantly, the *a* subunit of the F_o_ sector forms in combination with the *c* subunit, the proton channel that causes the rotation. The authors have detected a putative subunit *a*, but do not provide a detailed description of its structural features. It would be important to point out conserved features of this subunit that would form the channel, especially a conserved arginine residue that is critical for proton translocation. (Action to take: extend the Discussion section.)

---

## [Author Response]

Essential revisions:

*1) A manuscript currently deposited at BioRxiv (Salunke et al.;* https://www.biorxiv.org/content/early/2018/05/14/321620 *(now published in PLoS Biology, Salunke et al., 2018)) also describes subunit composition of Toxoplasma ATP synthase complex. In this manuscript, the authors find 20 novel subunits in addition to the 7 previously known subunits. In contrast, Huet et al. report just 11 novel subunits. Salunke et al. manuscript list 10 of the 11 subunits that Huet et al. report, which is quite reassuring, but the absence of 10 subunits listed by Salunke et al. in Huet et al. list is disconcerting. Huet et al. use an elaborate filtering approach to eliminate a large number of proteins detected through LC-MS/MS. The question thus arises as to whether the 10 additional subunits seen by Salunke et al. were filtered out by Huet et al. or that these were not detected in the first place. One possible reason for the discrepancy could be the tagged subunit used by the authors to pull down the complex for LC-MS/MS analysis: Huet et al. used the b subunit (present at 1:1 stoichiometry) whereas Salunke et al. used the β subunit (present at 3:1 stoichiometry). (Action to take: specify more clearly the stringency criteria applied to your analysis.)*

We were not aware of the study by Salunke et al. upon submission of our manuscript and therefore had no basis for the proposed comparison. In light of the recent publication, we have addressed the discrepancy between the two studies by including a new column in Supplementary file 1 indicating the name of each candidate protein according to the Salunke et al. paper. While our mass spectrometry analysis did detect all twenty ASAPs identified by Salunke et al., we excluded ten of them on the basis of lack of conservation between apicomplexan species. Our criteria are explicitly stated in the Results section under “A novel array of proteins is associated with the *T. gondii* ATP synthase” and within Supplementary file 1. We have also modified the Discussion section to include the following statement:

“During the revision of this manuscript, a similar set of proteins was identified as part of a proteomic analysis of the purified *T. gondii* ATP synthase complex (Salunke, Mourier, Banerjee, Pain, and Shanmugam, 2018). In addition to the novel subunits we described, the authors describe ten additional subunits, which our proteomic analysis identified, but we excluded on the basis of lack of conservation in either *C. muris, P. falciparum* or *C. velia* (Supplementary file 1). The novel subunits we propose may be closer to the core set of ATP synthase subunits present in alveolates; however, further studies will be needed to elucidate the function of most novel subunits regardless of their degree of conservation.”

2) The authors do not provide direct biochemical evidence for the proposed roles of ICAP2/18 in a functional ATP synthase complex. As proxy, oxygen consumption is measured in an ICAP2 knockdown mutant. This provides only an indirect measure as ATP synthase does not reduce oxygen. Further the seahorse data supporting this are incompletely reported as bar graphs. Glycolytic and mitochondrial contributions to ATP production in extracellular tachyzoites was characterized before (Lin et al., 2011) and mitochondria were found to contribute a small but quantifiable fraction to cellular ATP. The assay and conditions used in that paper glucose-free, 2-DG and glutamine supplemented should enable a more direct measurement of ATP in the ICAP2 knockdown mutant. (Action to take: to be addressed experimentally.)

To present the complete Seahorse data, we have included graphs for all experiments in Figure 6—figure supplement 1. However, we have retained the bar graphs as part of the main figure because these provide more concise and accessible results for these experiments.

We have conducted the proposed experiments based on the results of Lin et al. (2011) and used 2-DG, glucose, glutamine and oligomycin to measure the contribution of glycolysis and oxidative phosphorylation on ATP production by the cKD. The outcome of these experiments is shown in the new Figure 6C and demonstrates the importance of ICAP2 for ATP production from glutamine via oxidative phosphorylation. The Results section and Discussion section have been modified accordingly.

3) The authors are missing an important opportunity to point out implications of their findings in reference to the evolution of the apparent monophyletic clade Alveolata. The absence of the ATP synthase subunits they have found in Toxoplasma from the ciliates (and vice versa, i.e. ciliate ATP synthase subunits are absent in other alveolates) suggests almost a complete replacement of ATP synthase subunits at the branch point separating ciliates from the rest of the alveolate clade. Indeed, authors should include dinoflagellates in their comparative analysis (they actually have these "missing" subunits with orthology to Toxo, as pointed out by Salunke et al.). The chromoalveolate hypothesis has suggested monophyletic origin of all organisms possessing secondary red algal plastid and implying that ciliates may have lost their plastids. There are other studies that argue against this, and thus the present study provides further evidence against chromoalveolate hypothesis. One possibility is that the novel ATP synthase subunits shared by apicomplexans, chromerids (and dinoflagellates) may have originated from the nuclearly encoded genes of the secondary endosymbiont that targeted them to its mitochondrion, which now having been lost, got incorporated in the host nucleus for its own mitochondrion. (Action to take: extend the Discussion section.)

We appreciate the reviewers’ comments and have now included *Perkinsus* and *Symbiodinium* in our comparative analysis of ICAP2 and ICAP18. Figure 1—figure supplement 1 and Figure 2—figure supplement 1 have been modified accordingly and we now mention “other alveolates” in addition to apicomplexans and chromerids in discussing the conservation of these subunits. Although related subunits can be found outside the Apicomplexa, their conservation is incomplete and we can neither prove nor disprove the chromalveolate hypothesis on the basis of our findings. While ICAP2 and ICAP18 homologs can be found in dinoflagellates and perkinsids, we have not been able to find them in ciliates or outside of alveolates. In particular, the absence of these subunits in red algal genomes precludes us from drawing conclusions about their possible origin from the secondary endosymbiont, as the reviewers propose.

4) Importantly, the a subunit of the F_o_ sector forms in combination with the c subunit, the proton channel that causes the rotation. The authors have detected a putative subunit a, but do not provide a detailed description of its structural features. It would be important to point out conserved features of this subunit that would form the channel, especially a conserved arginine residue that is critical for proton translocation. (Action to take: extend the Discussion section.)

We have modified Figure 3 to include a diagram of the ATP synthase indicating the predicted position of the putative stator and *a* subunits. In addition, we have included an alignment of the putative *a* subunit in Figure 3—figure supplement 2 that indicates the position of the conserved arginine, and we have extended the Discussion section accordingly:

“In opisthokonts, the *a* subunit is a multi-span membrane protein with a conserved arginine residue critical for its function (Mitome et al., 2010). Topology predictions of the putative *a* subunit suggest the presence of three TM domains and an arginine residue that is conserved in apicomplexans and other alveolates (Figure 3—figure supplement 2).”